# *Fusarium* as a Novel Fungus for the Synthesis of Nanoparticles: Mechanism and Applications

**DOI:** 10.3390/jof7020139

**Published:** 2021-02-15

**Authors:** Mahendra Rai, Shital Bonde, Patrycja Golinska, Joanna Trzcińska-Wencel, Aniket Gade, Kamel A. Abd-Elsalam, Sudhir Shende, Swapnil Gaikwad, Avinash P. Ingle

**Affiliations:** 1Department of Biotechnology, Nanobiotechnology Laboratory, Sant Gadge Baba Amravati University, Amravati 444602, India; shitalbonde@gmail.com (S.B.); aniketgade@gmail.com (A.G.); sudhirsshende13884@gmail.com (S.S.); 2Department of Microbiology, Nicolaus Copernicus University, Lwowska, 87-100 Torun, Poland; golinska@umk.pl (P.G.); trzcinska@doktorant.umk.pl (J.T.-W.); 3Agricultural Research Center, Plant Pathology Research Institute, Giza 12619, Egypt; kamelabdelsalam@gmail.com; 4Academy of Biology and Biotechnology, Southern Federal University, 344006 Rostov-on-Don, Russia; 5Microbial Diversity Research Centre, Dr. D. Y. Patil Biotechnology and Bioinformatics Institute, Dr. D. Y. Patil Vidyapeeth (Deemed to be University), Tathawade, Pune 411033, India; gaikwad.swapnil1@gmail.com; 6Biotechnology Centre, Department of Agricultural Botany, Dr. Panjabrao Deshmukh Krishi Vidyapeeth, Akola, Maharashtra 444104, India; Ingleavinash14@gmail.com

**Keywords:** *Fusarium*, synthesis, nanoparticles, mechanism, medicine, agriculture, nanofactory, toxicity

## Abstract

Nanotechnology is a new and developing branch that has revolutionized the world by its applications in various fields including medicine and agriculture. In nanotechnology, nanoparticles play an important role in diagnostics, drug delivery, and therapy. The synthesis of nanoparticles by fungi is a novel, cost-effective and eco-friendly approach. Among fungi, *Fusarium* spp. play an important role in the synthesis of nanoparticles and can be considered as a nanofactory for the fabrication of nanoparticles. The synthesis of silver nanoparticles (AgNPs) from *Fusarium,* its mechanism and applications are discussed in this review. The synthesis of nanoparticles from *Fusarium* is the biogenic and green approach. Fusaria are found to be a versatile biological system with the ability to synthesize nanoparticles extracellularly. Different species of Fusaria have the potential to synthesise nanoparticles. Among these, *F. oxysporum* has demonstrated a high potential for the synthesis of AgNPs. It is hypothesised that NADH-dependent nitrate reductase enzyme secreted by *F. oxysporum* is responsible for the reduction of aqueous silver ions into AgNPs. The toxicity of nanoparticles depends upon the shape, size, surface charge, and the concentration used. The nanoparticles synthesised by different species of Fusaria can be used in medicine and agriculture.

## 1. Introduction

Nanotechnology is an emerging branch of science having enormous applications in almost all fields related to human life. It is mainly concerned with the synthesis and applications of materials having a size in the range of 1 to 100 nanometers [1]. Nanomaterials possess exceptionally novel properties such as a high surface-area-to-volume ratio, high reactivity, enhanced catalytic and biological properties. All these unique properties make the nanomaterials appropriate for a variety of applications including in biomedicine and agriculture [2,3,4]. To date, a variety of nanomaterials have been developed and many more are currently under investigation to be applied in biomedicine with the emphasis on various life-threatening diseases including cancer. Therefore, some precious metals (like silver, gold and platinum) and some magnetic oxides (i.e., magnetite Fe_3_O_4_) nanoparticles received much attention [5]. Similarly, various nanoparticles have been reported to have many beneficial applications in agriculture which mainly include plant growth promotion, usage as nanofertilizer, and nanopesticides [6].

It is well demonstrated that various nanoparticles can be synthesized using physical, chemical, and biological methods [7]. The conventional physical and chemical methods used for the synthesis of nanoparticles usually involve the usage of toxic chemicals and also generate waste, which can cause environmental pollution [8,9]. However, the synthesis of biogenic nanoparticles using different biological agents such as plants, microbes, and their products has gained considerable attention worldwide due to the rapid synthesis and their eco-friendly nature compared to physical and chemical methods.

Among the biological agents, fungi have been preferably used for the synthesis of a variety of nanoparticles. Synthesis of nanoparticles using fungi is referred as mycosynthesis [10,11,12,13] and is being dealt with under myconanotechnology [14]. Gade et al. [15] stated the advantages of using filamentous fungi over other biological agents (e.g., bacteria) for the synthesis of nanoparticles. These mainly include high tolerance towards heavy metals, it is easy to culture fungi at mass level, synthesis of nanoparticles is extracellular which reduces the cost of down streaming, etc.

To date, several fungi have been successfully exploited for the biological synthesis of nanoparticles, but from the available literature, it is evident that different species of *Fusarium* are the prime choice for scientists. Various species of *Fusarium* such as *Fusarium oxysporum*, *Fusarium semitectum*, *Fusarium acuminatum*, *Fusarium solani*, *Fusarium culmorum*, etc. and their different strains [16,17,18,19,20,21] have been used for the synthesis of nanoparticles like silver, gold, platinum, silica, palladium, etc. Several other fungal species could also be employed in nanoparticle synthesis as described in Mahmoud et al. [22], Elamawi et al. [23], Noor et al. [24], and many more. Considering these facts, in the present review, we have discussed the importance of *Fusarium* for biosynthesis of nanoparticles. Moreover, various other key aspects such as the mechanism of nanoparticle synthesis from *Fusarium* and their applications in biomedicine and agriculture and toxicity have also been discussed.

### 1.1. Diversity of Fusarium spp. for the Synthesis of Different Nanopartilces

Fungi play a very important role in solving major global problems for sustainable development as compared to other biological systems. They enhance resource efficiency, converting waste to valuable food and feed ingredients, making crop plants more robust to survive in climate change conditions, and functioning as host organisms for the production of new biological drugs [25]. Fungi are the most promising hotspots for finding new drug candidates, metabolites, and antimicrobials [8]. They are also responsible for the shift from chemical processes to biological processing, achieved by fungal enzymes instead of chemical processes in industries, such as textiles, leather, paper, and pulp, which have significantly helped to make the process eco-friendly by reducing the negative impact on the environment [16,26]. The fungal system has been found to be a versatile biological system with the ability to synthesize metal nanoparticles intracellularly as well as extracellularly. Moreover, they are preferred over other biological systems because of their ubiquitous distribution in nature, and therefore, many fungi have been explored for the production of various metal nanoparticles of different shapes and size. Out of diverse fungal genera used for the synthesis of nanoparticles, the genus *Fusarium* has been the choice of many investigators [13,16]. The advantages of using *Fusarium* spp. for the synthesis of nanoparticles are enlisted in Figure 1.

There are several reports on the synthesis of metal nanoparticles by different *Fusarium* spp. The exhaustive list of different *Fusarium* spp. involved with synthesis of different metal nanoparticles is given in Table 1.

### 1.2. F. oxysporum as a Novel Organism for Synthesis of Nanoparticles

Several fungi have been used for the biosynthesis of various nanoparticles as they exhibit many advantages over other biosystems. After directing to the mycosynthesis of nanoparticles especially from *Fusarium*, nanoparticles with better size and monodispersity could be achieved. Additionally, the extracellular production of enzymes has an added benefit in the downstream handling of biomass [62] as compared to other biosystems like bacteria and plants. Consequently, using these expedient properties of *Fusarium*, it could be comprehensively used for the rapid and eco-friendly biosynthesis of nanoparticles [53]. Gaikwad and colleagues [30] have screened eleven different *Fusarium* species isolated from various infected plant materials for the synthesis of silver nanoparticles (AgNPs). All the screened species revealed the ability for synthesis of AgNPs. Based on transmission electron microscopic (TEM) analysis, six *Fusarium* species—*viz. F. graminearum, F. solani, F. oxysporum, F. culmorum, F. scirpi, F. tricinctum*—synthesized smaller-sized particles, which signifies their prominence in AgNPs synthesis (Figure 2). Moreover, AgNPs synthesis from *F. scirpi*, *F. graminearum, F. tricinctum* was reported for the first time.

Khalil and coworkers [29] successfully synthesized and characterized AgNPs from *F. chlamydosporum* NG30 and *P. chrysogenum* NG85 which showed promising antifungal activity. The cellular mechanism of nanoparticle synthesis is yet to be completely understood; therefore, researchers have been trying to understand the mechanism at the cellular and molecular level [63]. They have reported the synthesis of gold nanoparticles (AuNPs) using *F. oxysporum* f. sp. *cubense* JT1 in 60 min. Naimi-Shamel et al. [64] also proved that *F. oxysporum* has benefits like fast growth rate, low-cost biomass management, safety and easy processing for synthesis of AuNPs. AuNPs synthesized from *F. oxysporum* are found to have a high tendency of conjugation with β-Lactam antibiotics, and this affinity makes them a better detoxification agent as well in various areas including medicine [65]. An endophytic strain of *F. solani* isolated from *Chonemorpha fragrans* plant was used to synthesise nanoparticles with anticancer activity [60]. El-Sayed and El-Sayed [66] synthesized silver, copper and zinc biocidal nanoparticles from *F. solani* to combat multidrug-resistant pathogens. *F. oxysporum*-mediated AgNPs surface coated with different proteins and biomolecules act as a potential antimicrobial agent and proved by protein–ligand interaction in silico studies [66]. In another study by Birla et al., *F. oxysporum* produced more protein at an optimized temperature between 60° and 80 °C which showed the progressive increase in the rate of nanoparticle synthesis [33].

## 2. Mechanism of Nanoparticle Synthesis from *Fusarium*

As discussed earlier, members of the genus *Fusarium* can synthesize metal nanoparticles both intracellularly and extracellularly. As far as the mechanism of extracellular mycosynthesis is concerned, it is proposed that metabolites such as enzymes, proteins, polysaccharides, flavonoids, alkaloids, phenolic and organic acids, etc. secreted by fungus-like *Fusarium* for their survival when exposed to different environmental stresses are mostly responsible for the reduction of metals ions to metallic nanoparticles through the catalytic effect [67]. Moreover, the same metabolites act as reducing and stabilizing agents which are further responsible for the growth and stabilization of biogenic metal nanoparticles [13,22]. Figure 3 represents the schematic illustration of the general mechanism involved in the synthesis, growth, and stabilization of metal nanoparticles using fungus such as *Fusarium*.

One of the hypothetical mechanisms proposed is that NADH-dependent nitrate reductase enzyme secreted by *F. oxysporum* is responsible for the reduction of aqueous silver ions into AgNPs [16]. A similar mechanism has been proposed by Ingle et al. [10] in case of synthesis of AgNPs from *F. acuminatum*, and they also pointed the involvement of cofactor NADH and nitrate reductase enzyme in the biosynthesis of AgNPs because they reported the presence of nitrate reductase in fungal cell-free extract using specific substrate utilizing discs for nitrate purchased from Hi-Media Pvt. Ltd. Mumbai, India.

In addition, Duran et al. [68] and Kumar et al. [69] proposed almost similar mechanisms for the biosynthesis of AgNPs from *F. oxysporum*. In the former study, the authors reported the role of anthraquinone and the NADPH-nitrate reductase in the biosynthesis of AgNPs, and it was hypothesized that the electron required to fulfill the deficiency of aqueous silver ions (Ag^+^) and convert it into Ag neutral (Ag^0^ i.e., AgNPs) was donated by both quinone and NADPH. However, in the later study, it was demonstrated that the reduction of NADPH to NADP^+^ and the hydroxyquinoline possibly acts as an electron shuttle transferring the electron generated during the reduction of nitrate to Ag^+^ ions, converting them to Ag^0^ (Figure 4).

There are reports suggesting various hydroquinones to act as electron shuttles reducing the metal ions. *F. oxysporum* f. sp. *cubense* JT1 demonstrated to have the capacity to reduce the gold ions to AuNPs [48]. Moreover, as per Ahmad et al. [16], the capacity of reducing metal ions is species-specific. The reductase specific to *F. oxysporum* and *F. moniliforme* were not able to synthesize AgNPs intracellularly and extracellularly. In addition to extracellular mechanisms, there are few mechanisms proposed for intracellular mycosynthesis of metal nanoparticles. In the case of *Fusarium*-mediated intracellular mycosynthesis, metal nanoparticles usually formed below the cell surface and this may be due to the reduction of metal ions by metabolites (i.e., enzyme) present in the cell membrane. Generally, a two-step mechanism has been proposed for intracellular mycosynthesis of nanoparticles. In the first step, aqueous metal ions are attached to the fungal cell surface by the electrostatic interaction between lysine residues and metal ions (M^+^). However, in the second step, the actual mycosynthesis of nanoparticles occurs by the enzymatic reduction of metal ions (M^0^), which leads to the aggregation and formation of nanoparticles [49] (Figure 5).

Moreover, various other studies performed on mycosynthesis proposed the role of different other enzymes and proteins. However, among all these mechanisms for extracellular mycosynthesis, the hypothetical mechanism involving the role of NADH-dependent nitrate reductase enzyme has been widely accepted.

### 2.1. Biomedical Applications of Nanoparticles Synthesized Using Fusarium Spp.

Biomedical application is an expanding field of research with tremendous prospects for the improvement of the diagnosis and treatment of human diseases [70]. The dispersed nanoparticles are usually employed in nanobiomedicine as fluorescent biological labels [71], as well as drug and gene delivery agents [72].

The biologically synthesized AgNPs could have many applications in areas such as non-linear optics, spectrally selective coating for solar energy absorption and intercalation materials for electrical batteries, as optical receptors, catalysis in chemical reactions, bio labelling [73], and as antibacterial agents [74,75,76]. The biomedical applications of *Fusarium*-mediated synthesized nanoparticles are shown in Figure 6.

#### 2.1.1. Antibacterial Activity of Nanoparticles

Gupta and Chundawat [41] used *Fusarium oxysporum* for the production of platinum nanoparticles. The zones of inhibition against microbes were studied by the agar well diffusion and agar dilution methods. It helps to know the minimum inhibitory concentration of platinum NPs. The minimum inhibitory concentration of platinum NPs was found to be 62.5 µg ml^−1^ for *E. coli*, which is relatively better than that of commercially available drug ampicillin. Also, the antioxidant activity was studied by the α, α-diphenyl-β-picrylhydrazyl (DPPH) method and platinum nanoparticles showed 79% scavenging activity. Duran and his co-workers [68] synthesised AgNPs by *Fusarium oxysporum* and integrated into the textile fabric for the inhibition of bacterial contamination such as *Staphylococcus aureus.*

The uropathogenic *Escherichia coli* (UPEC) form biofilms. The prevalence of urinary tract infections (UTIs) is due to the inaccessibility of the antibiotics into the highly complex structure of the biofilm. However, with the appearance of antibiotic multi-resistant UPEC strains, alternatives to the treatment of UTIs are fewer. AgNPs are an effective treatment of UPEC infections due to its physicochemical properties that confer them antibacterial activity against biofilm structured cells [54].

#### 2.1.2. Antiviral Activity of Nanoparticles

The interaction between AgNPs and viruses is attracting pronounced interest due to the potential antiviral activity of these particles. Elechiguerra et al. [77] reported that the smaller AgNPs are capable of reducing viral infectivity by inhibiting attachment to the host cells. It has been established that AgNPs undergo a size-dependent interaction with herpes simplex virus types 1 and 2, and with human parainfluenza virus type 3. Also, the authors confirmed that smaller nanoparticles were able to decrease the infectivity of the viruses [78]. Gaikwad et al. [30] synthesized AgNPs using *Fusarium oxysporum* and other fungi, which showed potential for reducing the replication of HSV-1, HSV-2, and HPIV-3 in cell cultures. The AgNPs formed by *F. oxysporum* were the most effective and presented low cytotoxicity [9].

#### 2.1.3. Anticancer Activity of Nanoparticles

Husseiny et al. [79] reported the antibacterial and antitumor potential of AgNPs synthesized from *Fusarium oxysporum*. The nanoparticles were effective to inhibit *E. coli* and *S. aureus*, and also a tumor cell line. A low IC50 value (121.23 µg cm^−3^) for MCF-7 cells (human breast adenocarcinoma) was gained following exposure of the cells to the nanoparticles, indicating high cytotoxicity and the potential for tumor control. The effect was recognized by the involvement of the AgNPs in the disruption of the mitochondrial respiratory chain, which led to the production of reactive oxygen species and hindered the production of adenosine triphosphate (ATP), consequently damaging the nucleic acids.

Clarance et al. [60] explored the anticancer potential of the AuNPs obtained by the green synthesis method using an endophytic strain *Fusarium solani* ATLOY—8 isolated from *Chonemorpha fragrans*. The AuNPs were tested for their cytotoxicity on cervical cancer cells (He La) and human breast cancer cells (MCF-7); the NPs exhibited dose-dependent cytotoxic effects. The results delivered an apparent and versatile biomedical application for a safer chemotherapeutic agent with little systemic toxicity.

#### 2.1.4. Antifungal Activity of NPs

Bansod et al. [34] reported bioconjugate-nano-PCR as a rapid and specific method for the identification of *Candida* species in less time. The DNA sample of *Candida albicans* was conjugated with AuNPs and AgNPs synthesized from *F. oxysporum*. The use of this nanoparticle-altered template enhances the sensitivity and specificity of the traditional PCR assay. It is helpful in molecular diagnostics and therapeutics. It is demonstrated as an effective method for the identification of *Candida* sp. with a low concentration of DNA and less time. In another study, it was demonstrated that AgNPs synthesized by *F. oxysporum* were inhibitory to pathogenic fungi such as *Candida* and *Cryptococcus* [76].

Horky et al. [80] summarized the current findings of mycotoxins and their elimination by nanoparticles. They concluded that nanomaterials have interesting adsorption properties, which make them promising for mycotoxin elimination.

#### 2.1.5. Antiparasitic Activity Against Vectors

Some studies showed that AgNPs synthesized from different fungal species can be employed in pest control [81]. In a study, Dhanasekaran and Thangaraj [82] evaluated the larvicidal activity of biogenically synthesized AgNPs against the larvae of *Culex* mosquito vector, which causes filariasis. The authors reported that 5 mg/L AgNPs were responsible for 100% mortality of the Culex larvae. This research has opened a new area of research that the biogenically synthesized AgNPs can be used for the control of mosquito vectors causing diseases like filaria, malaria. dengue, etc.

## 3. Applications in Agriculture

Agriculture plays a major role in the economy as it is the backbone of most developing countries. The worldwide population is growing day-by-day very rapidly and it is predicted that it will reach about eight billion by 2025 and 9.6 billion by 2050. It is widely recognized that global agricultural productivity must increase to feed a rapidly growing world population [83,84,85]. For the improvement in crop productivity, nanotechnology provides new agrochemical agents and new delivery mechanisms, and it promises to reduce pesticide use as NPs could be used in the variable applications concerned with agriculture. The applications of nanotechnology (as shown in Figure 7) can boost agricultural production, which includes the nano-formulations of agrochemicals to use as pesticides and fertilizers for crop improvement, nano-biosensors in crop protection for the identification of diseases and residues of agrochemicals, and nano-devices for the genetic manipulation of plants, etc. In agriculture, nanobiotechnology is used to improve the food production, with corresponding or even higher values of nutrition, quality and safety. Efficient application of pesticides, fertilizers, herbicides and plant growth regulators is the very critical way to get better crop production [86,87]. Nanocarriers could be used to achieve the controlled release of herbicides, pesticides, and other plant growth regulators. For example, poly (epsilon-caprolactone) nanocapsules have been recently developed as a herbicide carrier for atrazine [88]. The mustard plants (*Brassica juncea*) when treated with atrazine-loaded poly (epsilon-caprolactone) nanocapsules more significantly boosted the herbicidal activity than that of commercial atrazine, demonstrating a drastic decline in net photosynthetic rates. Moreover, the stomatal conductance and oxidative stress increased considerably, which ultimately reduced the weight and growth of the plants [88]. Likewise, other nanocarriers such as silica nanoparticles [89] as well as polymeric nanoparticles [43] have also been developed for delivering the pesticides in a prescribed manner. Nanocarriers could be employed to perfectly achieve the delivery as well as the slow release of these species, which is known as “precision farming”. This helps to improve the crop yield without damaging the soil and water [90].

Most significantly, the application of nanoencapsulation could lower down the herbicide dosage leading to a safer environment. In addition to nanocarriers, NP-mediated gene transfer in plants was employed for the development of insect-resistant crop varieties. The detailed account for the gene or DNA transfer could be established in earlier available reviews [91,92]. Al-Askar et al. [75] demonstrated that AgNPs biosynthesized by *F. solani* isolated from wheat were shown to be effective for the treatment of wheat, barley, and maize seeds contaminated by different species of phytopathogenic fungi. Moreover, metal oxide nanomaterials such as CuO, TiO_2_, and ZnO are extensively studied for plant protection from pathogen infections because of their intrinsic toxicity. For example, ZnONPs efficiently inhibited fungal growth such as *F. graminearum* [93], *Aspergillus flavus*, *Aspergillus fumigatus*, *Aspergillus niger*, *F. culmorum* and *F. oxysporum* [94]. The use of CuNPs as an antimicrobial agent against plant pathogens has been reported in several publications [95,96]. Mineral fertilizers used conventionally undergo substantially high losses besides lower uptake efficiencies of the nutrients. Those economic losses will be overcome by the development of nanofertilizers, which could be the novel solution. Nanofertilizers can reduce the nutrient loss as well as increasing nutrient adaptation by soil microbes and crops [97]. Nanofertilizers are mainly the micro-nutrients at nanoscale for Mn, Cu, Fe, Zn, Mo, N, and B, and are commercialized and available under different brand names in the market such as Nano-Ag Answer^®^, NanoPro™, NanoRise™, NanoGro™, NanoPhos™, NanoK™, NanoPack™, NanoStress™, NanoZn™, pH5^®^, etc. [87]. The use of other nanomaterials as an alternative for the typical conventional crop fertilizers, such as carbon nano-onions [98] and chitosan nanoparticles [99], was noted to boost the crop growth and quality. Shende et al. [100] reported the plant growth-promoting activity of biogenic CuNPs on pigeon pea (*Cajanus cajan* L.) crops that suggested the use of these nanoparticles as a nanofertilizer for the development of sustainable agriculture. It is estimated that the novel nanofertilizers will encourage and makeover current fertilizer production industries in the next decade [87].

Because of several advantageous characteristics of nanomaterials, nanosensors, particularly wireless nanosensors, have also been developed to monitor nutrient efficiency in crop plants, crop diseases, and growth, along with the environmental conditions in the field. Particularly, engineered nanosensors are capable of detecting chemicals like pesticides, herbicides, and pathogens at trace amounts in food and agricultural systems. Such an in situ and real-time monitoring system facilitates the remediation of probable crop losses as well as perking up the crop production, accompanied by the suitable application of nanopesticides, nanoherbicides, and nanofertilizers. Abbacia et al. [101] reported that the copper-doped montmorillonite will possibly be used for on-line monitoring of propineb fungicide in an aquatic environment i.e., in both fresh and salty water, with a low detection limit of about 1 mM [101]. In another study, it was demonstrated that the nanomaterials such as graphene could be developed for the detection of the pathogen in wastewater [102] and purification of drinking water can be carried out [103], signifying its potential application in aquaculture. Moreover, various other nanomaterials like carbon nanotube [104], CuNPs [105], AgNPs [106], and AuNPs [107] can be used as nanosensors designed for the real-time monitoring of crop health and growth along with the environmental conditions in the field. Table 2 showed different *Fusarium* isolates used in the green synthesis of nanoparticles.

## 4. Toxicity of *Fusarium* Nanoparticles

Nanoparticles are unique materials as they have property combinations compared with conventional materials [114]. There is a wide range of applications of NPs such as in human health appliances, industrial, medical and biomedical fields, engineering, electronics, and environmental applications [115]. Among all nanomaterials, AgNPs are the most widely used in medicine, medicinal devices, pharmacology, biotechnology, electronics, engineering, energy, magnetic fields, and also in environmental remediation [116]. Their highly effective antibacterial activity has found applications in industrial sectors including textiles, food, consumer products, medicine, etc. [117].

The unique physical, chemical and biological (e.g., antimicrobial, anticancer, antiparasitic) properties of nanoparticles differ largely from corresponding bulk materials and make them a high-demand material in different sectors. However, the widespread and increased use of nanoparticles may pose a risk to both the environment and living organisms by increasing the level of toxicity [118]. To date, several studies have used different model cell lines to exhibit the cytotoxicity of nanoparticles from *Fusarium* species, mainly AgNPs, and their underlying molecular mechanisms [31,60,65,119]. Biogenic nanoparticles are capped with natural molecules like proteins [120,121,122]. This capping is defined as corona. This nanoparticle corona significantly affects the biological response [123]. Based on the surface affinity and exchange rate, the corona can be divided into two forms: hard corona and soft corona. The soft corona proteins are ’vehicles’ for the silver ions, whereas the hard coronas are rigid for the trespass into the cellular system [124]. The functional groups of the corona play a key role in the formation of the nanoparticle–protein corona [125]. These functional groups along with the protein charges also regulate the cytotoxic properties of the nanoparticle corona [123,125]. The surface charge of nanoparticles plays an important role in their bactericidal activity against both Gram-positive and Gram-negative bacteria, as exemplified by AgNPs [126]. It was confirmed using transmission electron microscopy (TEM) that AgNPs can penetrate cellular compartments such as endosomes, lysosomes, and mitochondria [127]. Several reports indicate that the proteins contained in the nanoparticle corona interact with the cells and not the nanoparticles themselves [128,129]. Thus, the corona formation and composition have important implications for both toxicity [130] and internalization [131].

To sum up, particle size [132], particle shape [133], particle surface properties [134], biological fluid properties, and composition affect the corona structure and thus the adverse effects on human health and the environment [131,135].

### 4.1. Effect of Size and Shape of Nanoparticles on Cytotoxicity

It is claimed that apart from the size, the shape of nanoparticles affects their toxicity to cells. Mohamed et al. [31] investigated the cytotoxicity of the two different shapes of zinc oxide nanoparticles (ZnONPs) biosynthesized from *Fusarium keratoplasticum* (A1-3) and *Aspergillus niger* (G3-1). These nanoparticles displayed a similar size (10–42 and 8–38 nm) but different shapes, namely hexagonal and nanorods, respectively. It was reported that a safe dose of these nanoparticles for applications in animal cells should be lower than 20.1 and 57.6 ppm, respectively. Therefore, the rod ZnONPs were more applicable to safety at high concentrations in contrast to hexagonal ZnONPs [31]. Soleimani and co-authors [136] studied biological activity of different shapes (cube, sphere, rice and rod) of AgNPs synthesized using chitosan in acetic acid solution and 0.2 M of AgNO_3_ (cubes) and *F. oxysporum,* starch and 0.08 M of AgNO_3_ at pH 6.8 (spheric) or 1.0 M of AgNO_3_ at pH 3.0 (nanorice), and finally, *F. oxysporum*, starch and 1.2 M AgNO_3_ at pH 3.0 and 30 °C for 3 days (blunt ends rods) or with 10-days incubation (sharp-ends rods). They found that silver nanostructures with different shapes are not inherently toxic to human cells at concentrations lower than 10 µg ml^−1^, whereas for higher concentrations cell viability decreased in a shape and dose-dependent manner. Nanocubes, nanorice and sharp-nanorods were found to be more toxic than spheres and blunt-nanorods. The former significantly decreased cell viability at concentrations 25 µg mL^−1^ or higher, and the latter at concentrations of 50 and 75 µg mL^−1^. Moreover, they showed that cubic nanoparticles inhibited the growth of all tested bacteria (*Bacillus subtilis, Staphylococcus aureus, Escherichia coli,* and *Pseudomonas aeruginosa*) at the lowest-used concentrations (10 ppm). The toxic effect of nanostructures against bacterial cells increased in order spheres, blunt-rods, nanorice, sharp-rods, and nanocubes [136]. It was concluded that due to stronger vertex in sharp ends of silver nanostructures, some geometries of AgNPs, especially cubic structures, have more interaction with the bacterial cell membrane, resulting in stronger biological activity [136]

### 4.2. Mechanisms of Nanoparticle Cytotoxicity

The increased use of AgNPs also increases the concentration of silver ions in soil and water that can adversely affect human and animal health or the environment [137,138].

Little is known about the diversified mechanisms of the action of nanoparticle cytotoxicity, as well as their short- or long-term exposure outcomes on human physiology [139,140]. Many authors claim that the silver ions released from AgNPs through the surface oxidation are the main mechanism that induces cytotoxicity, genotoxicity, immunological responses in biological systems and even cell death [141,142,143,144,145].

Several recent studies revealed that cytotoxicity of AgNPs occur due to the minimum release of silver ions [146,147,148]. Sambale et al. [149] showed that the mechanism of cell death is different when cells are cultivated with silver particles or ions. Cells treated with particles suffered apoptosis while those cultivated with ions die due to necrosis. They also reported that AgNPs that translocate into cytoplasm through diffusion or channel proteins are oxidized by cytoplasmic enzymes, thereby releasing silver ions. These ions may interact with thiol groups of mitochondrial membrane proteins, causing mitochondrial dysfunction and generating reactive oxygen species (ROS) production [149].

### 4.3. Effect of Nanoparticles on Cell Membranes

Nanosilver can interact with cellular membranes and cause toxicity. In particular, nanosilver can interact with the bacterial membrane and this is considered the main mechanism for the antimicrobial toxicity of nanosilver. The AgNPs damaged and destroyed bacterial cells by penetrating and accumulating in the bacterial membrane [150,151,152]. Khan et al. [153] studied the interaction of nanosilver with five types of bacteria. They found that the adsorption of nanosilver on the bacterial surface or interaction with extracellular proteins is dependent on pH, zeta potential, and NaCl concentration. Baruwati et al. [154] studied “green” synthesized nanosilver from Chinese green tea (*Camellia sinensis*) and found that exposure to these particles alters the membrane permeability of barrier cells (intestinal, brain endothelial) and stimulates oxidative stress pathways in neurons [154].

Metal nanoparticles (copper, silver and zinc oxide) from *Fusarium solani* KJ 623702 were tested against multidrug-resistant bacteria (*P. aeruginosa* and *S. aureus*), *E. coli*, *Klebsiella pneumoniae* and *Entreococcus* sp. and mycotoxigenic *Aspergillus awamori, A. fumigatus* and *F. oxysporum*. The antimicrobial activity of metal nanoparticles increased as follows: CuNPs, ZnONPs, and AgNPs. Besides the internalization of AgNPs with the cell wall resulting in the presence of pits, the fragmentation, complete disappearance of cellular contents, disorganization and leakage of internal components were observed after treatment of microorganisms with mycogenic AgNPs. Authors claimed that AgNPs attach to the microbial cell membrane and alter its structure, transport activity, penetrability, prompt neutralization of the surface electric charge and produce cracks and pits through which internal cell contents are effluxed [44].

Moreover, AgNPs synthesized using *F. oxysporum* filtrate containing reductase enzymes and quinones showed a toxic effect on the human liver cell line (Huh-7). The IC_50_ values were found to be 11.12 µM for propidium iodide assay which is an intercalating fluorescent dye and does not permeant to live cells [155]. Similarly, cell viability (hamster lung fibroblasts, V79) after the treatment with AgNPs synthesized from *F. oxysporum* was studied using neutral red uptake (NRU) assay [156]. These small (around 8 nm) nanoparticles showed a non-cytotoxic effect on fibroblasts at concentrations until 16 µM. The IC_50_ of mycogenic AgNPs was recorded at a concentration of 22 µM [156].

Vijayan et al. [157] estimated the release of haemoglobin from red blood cells due to the disruption of the cell membrane by the AgNPs from *F. oxysporum*. The percentage of haemolysis increased from 3.8 to 32.0 with an increase in AgNPs concentration from 25 to 150 µg ml^−1^ (at 25 µg ml^−1^ interval unit). The authors concluded that these biogenic AgNPs are safe to use as a drug because at safe concentrations they showed antibacterial effect against *Escherichia coli*, *Klebsiella pneumoniae*, *Pseudomonas aeruginosa* and *Salmonella typhi* [157].

### 4.4. Effect of Nanoparticles on Mitochondria and/or Metabolic Activity

Nanoparticles by mitochondrial membrane damage disturb respiratory chain activity and ATP synthesis may generate ROS production, leading to oxidative stress and eventually apoptosis [158,159]. Many authors reported the effect of metal (Ag and Au) and non-metal (selenium) nanoparticles from *Fusarium* species on the viability of human or animal cells on the basement of activity of cell/mitochondrial enzymes using mainly spectrophotometrical MTT (3-(4,5-dimethylthiazol-2-yl)-2,5-diphenyltetrazolium bromide) assay [155,156,160,161].

The toxicity of AuNPs produced by *F. oxysporum* with sizes of around 20 to 50 nm was measured by MTT assay [161]. The biologically produced AuNPs had a dose-dependent toxic effect on the human fibroblast cell line (CIRC-HLF). The IC_50_ value of the NPs was determined as 2.5 mg mL^−1^ [161]. The AuNPs from *Fusarium solani* ATLOY-8 strain were studied against cancer cell lines, namely HeLa na MCF-7 (breast cancer cells) and the human embryonic kidney (HEK) cell line using MTT assay. The anticancer potential of NPs against both cancer cell lines (IC_50_ values of 1.3 and 0.8 mg mL^−1^, respectively) and their insignificant activity on the HEK cell line was reported [60].

AgNPs from *Fusarium oxysporum* were studied on mouse fibroblasts (3T3), hamster lung fibroblasts (V79 line), and human liver (Huh-7) cells using MTT and calcein assays [155,156,161]. AgNPs showed dose-dependent toxic effects on the mouse fibroblast. IC_50_ of mycogenic nanoparticles using MTT assay were found to be 0.312 mg mL^−1^ (= 1.187 ppm mL^1^). The applied concentrations of the silver NPs equal to 0.625 mg mL^−1^ (2.375 ppm mL^−1^) was determined as the toxic dose [60]. Marcato and coauthors [156] reported that AgNPs synthesized using *Fusarium oxysporum* showed a non-cytotoxic effect toward the V79 fibroblast cell line until 16 µM, evaluated by MTT assays with an IC_50_ of 22 µM; this was similar in NRU assay. AgNPs synthesized using *F. oxysporum* filtrate demonstrated a comparable dose-response relationship and similar IC_50_ values of 10.61 µM for MTT and 9.10 µM for calcein assay [155]. Small AgNPs (5–13 nm) produced by *F. oxysporum* were also found to be toxic to the human breast carcinoma cell line (MCF-7). The IC_50_ value of these NPs was found to be 121.23 µg mL^−1^ [78].

Another species of the genus *Fusarium*, namely *F. semitectum*, was used for the synthesis of selenium nanoparticles (SeNPs) which were tested for their anticancer potential on Caco-2 human colon cancer, A431 skin cancer, and SNU16 stomach gastric cancer cell lines, as well as toward THLE2 normal liver and Vero normal kidney cell lines [57]. These biogenic SeNPs showed anticancer potential toward colon, skin, and stomach gastric cancer cells (IC_50_ of 10.24, 13.27 and 20.44 µg mL^−1^, respectively), no cytotoxic effects on normal liver cells, and weak toxicity on normal kidney cells [57].

### 4.5. Effect of Nanoparticles on Cell Proteins

Many authors have reported that silver ions and AgNPs can interact with various chemical groups, including sulfide and chloride [162,163]. Thiol molecules are found to be conjugated to several membrane proteins in the cell membrane, cytoplasm and mitochondria, which may serve as targets for AgNPs or Ag^+^ ions [164]. AgNPs also bind to thiol groups in enzymes, such as NADH dehydrogenase, and disrupt the respiratory chain, finally generating ROS. As shown in studies by Soleimani et al. [136] the cubic AgNPs from *F. oxysporum* at a concentration of 30 ppm not only caused a toxic effect in MCF-7 cells and inhibited growth of the Gram-positive and Gram-negative bacteria, but also completely degraded proteins (albumin), whereas other shaped nanoparticles (rods, rice and spheres) had no denaturing effect on protein structure. They concluded that cubic nanosilver displays stronger biological activity when compared to rod and spherical AgNPs [136].

### 4.6. Effect of Nanoparticles on Cell Nucleic Acids

AgNPs, especially those smaller than 10 nm, have been shown to diffuse through the nuclear pores into the nucleus, causing DNA damage, chromosomal aberrations, and cell cycle arrest, resulting in genotoxicity in human cell lines (e.g., fibroblasts and glioblastoma cells) [162,163].

Clarance et al. [60] studied the effect of mycogenic AuNPs on cancer cells using dual AO/EtBr (acridin orange/ethidium bromide) staining. The cell line appeared as a dense red colour after treatment with nanogold, which was mainly due to apoptotic cell death. The DAPI staining revealed nuclear fragmentation and condensation in MCF-7 cells treated with AuNPs. It is well-proven that when cells undergo an apoptotic death, their DNA becomes dense, fragmented and with condensed chromatin [165,166].

It was also reported that the interaction of AgNPs synthesized from *Fusarium mangiferae* with DNA in *Staphylococcus aureus* cells resulted in an enormous reduction or degradation of both chromosomal and plasmid DNA in bacterial cells when compared with untreated cells. This gel electrophoresis study of AgNPs-treated nucleic acids also revealed that the RNA smearing was immensely decreased. These results suggest that AgNPs may prevent DNA replication via binding, and lead to cell death [167].

Spherical AgNPs synthesized by using aqueous extracts of green *Calligonum comosum* stem and *Fusarium* sp. (mean size of 105.8 and 228.4 nm, respectively) were tested for genotoxicity in bacterial cells (*S. aureus*). The agarose gel electrophoresis and (UV)–Vis spectrophotometer were used to evaluate the antimicrobial activity of AgNPs by their influence on DNA. A low concentration of DNA was observed in bacterial cells treated with plant-synthesized and mycosynthesized AgNPs when compared to untreated cells. Based on the obtained results, it was concluded that the smaller-sized nanoparticles diffuse more easily than the larger ones, which partially explained the higher toxicity of plant-synthesized than myco-synthesized nanoparticles on *S. aureus* cells. The destruction of the membrane integrity or inhibition of DNA replication were proposed by authors as AgNPs mechanisms of antibacterial action [58].

### 4.7. Other Nanoparticle Activity

Salaheldin et al. [119] reported that AgNPs synthesized by *F. oxysporum* caused remarkable vacuolation in the human breast carcinoma cell line (MCF-7), thus indicating potent cytotoxic activity. Al-Sharqi [168] tested these mycogenic AgNPs on mice for 21 days and showed dilation in the collecting tubule and dilation in the renal corpuscle with haemorrhage in the interstitial space between the tubules. Based on performed studies, the authors conclude that AgNPs can enter and translocate within the cell, and that the size of the AgNPs varies with its toxic effects on the cell and the cell organelles. Therefore, it was assumed that all nanoparticles are toxic and most likely only free nanoparticles that can penetrate small organelles such as mitochondria may trigger adverse health effects [168].

## 5. Conclusions

The potential of applications of nanotechnology particularly as nanomedicine and in agriculture has generated revolution. The bioinspired synthesis of nanoparticles using fungi is a novel and emerging field of green and sustainable nanotechnology. Among these, *Fusarium* spp. are the most studied fungi for the biosynthesis of nanoparticles. The process is green, eco-friendly, and economically viable. The applications of different nanoparticles as antimicrobials, and antiparasitic and anticancer agents have attracted the interest of the researchers globally. Moreover, these biogenic nanoparticles, synthesized by Fusaria, have huge potential in agriculture for plant growth promotion, as nanofertilisers and as fungicidal agents of the new generation. However, the mechanism of the synthesis of nanoparticles remains unclear and warrants further studies to unravel the mechanism. Looking at the potential applications of Fusarium-mediated nanoparticles, the toxicity is a major issue that depends upon shape, size, surface charge, and the dose of nanoparticles used. Thus, it can be recommended that *Fusarium* is a promising and novel fungus for the biosynthesis of nanoparticles and its potential biomedical and agricultural applications.

## Figures and Tables

**Figure 1 jof-07-00139-f001:**
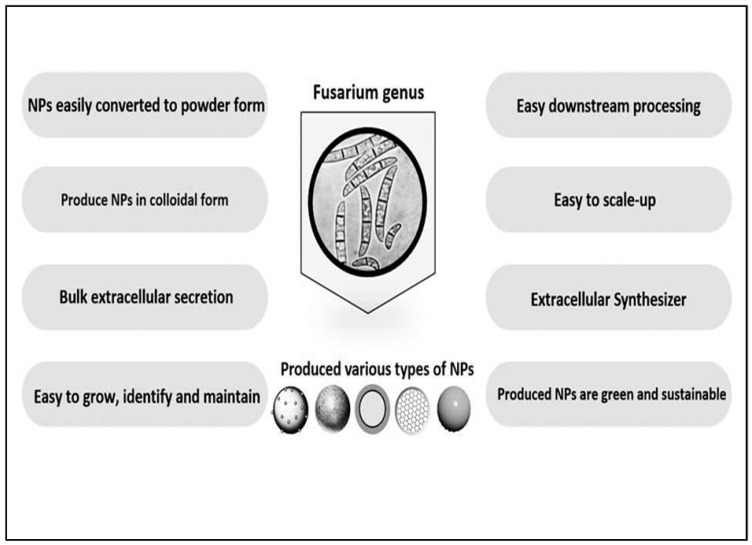
The advantages of using *Fusarium* spp. for the synthesis nanoparticles (NPs).

**Figure 2 jof-07-00139-f002:**
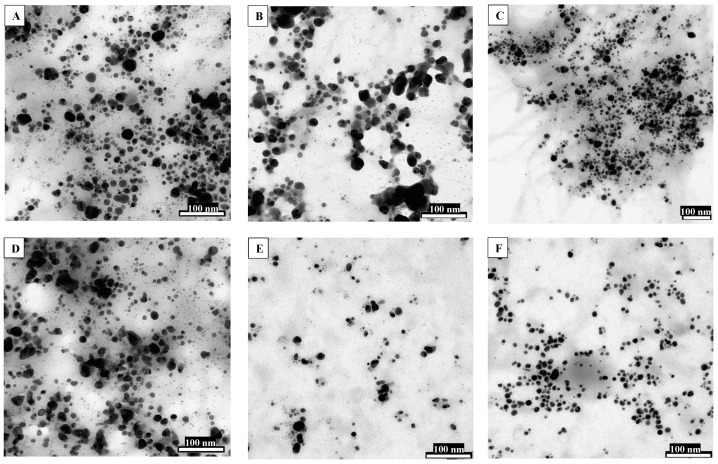
TEM micrographs presenting sphere-shaped AgNPs synthesized by various *Fusarium* species. (**A**) *F. graminearum;* (**B**) *F. solani;* (**C**) *F. oxysporum;* (**D**) *F. culmorum;* (**E**) *F. scirpi;* (**F**) *F. tricinctum*. (Reproduced with permission from Gaikwad et al. [30]).

**Figure 3 jof-07-00139-f003:**
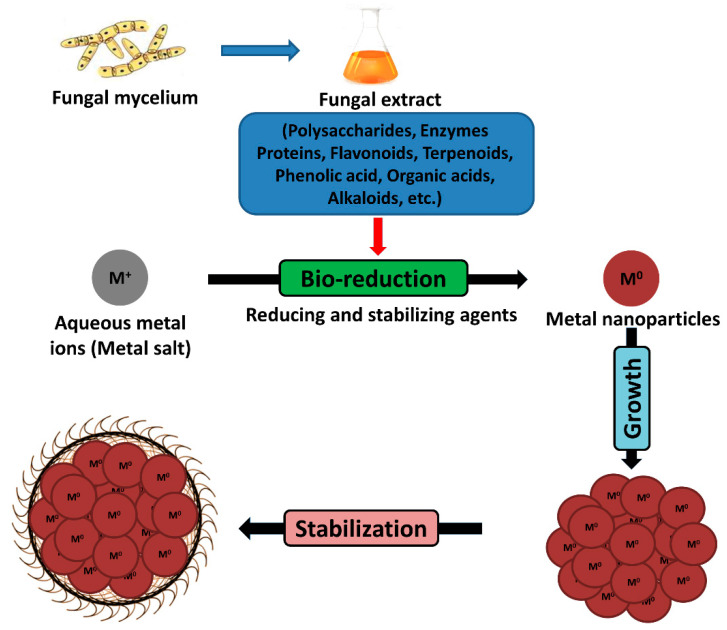
General mechanism involved in the synthesis, growth, and stabilization of metal nanoparticles using fungus.

**Figure 4 jof-07-00139-f004:**
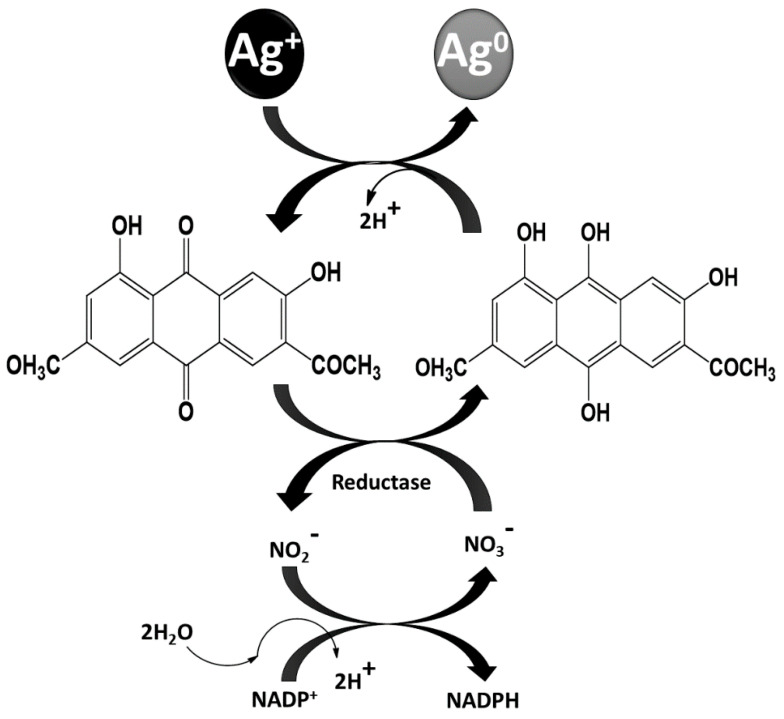
Hypothetical mechanism of AgNPs biosynthesis from *F. oxysporum* (Adapted from Duran et al. [67], an open-access article).

**Figure 5 jof-07-00139-f005:**
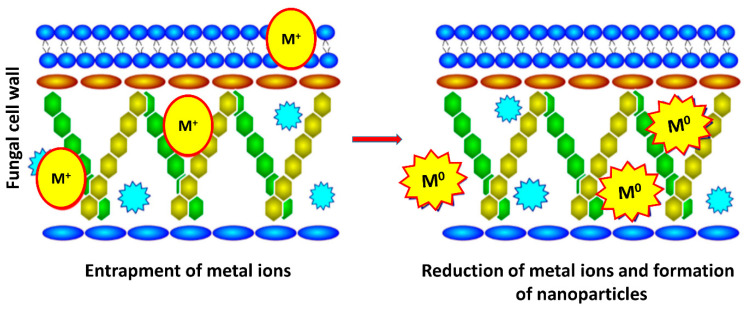
Hypothetical intracellular mechanisms for the synthesis of metal nanoparticles from *F. oxysporum* (Adapted and modified from Yadav et al. [13]; with copyright permission from Springer).

**Figure 6 jof-07-00139-f006:**
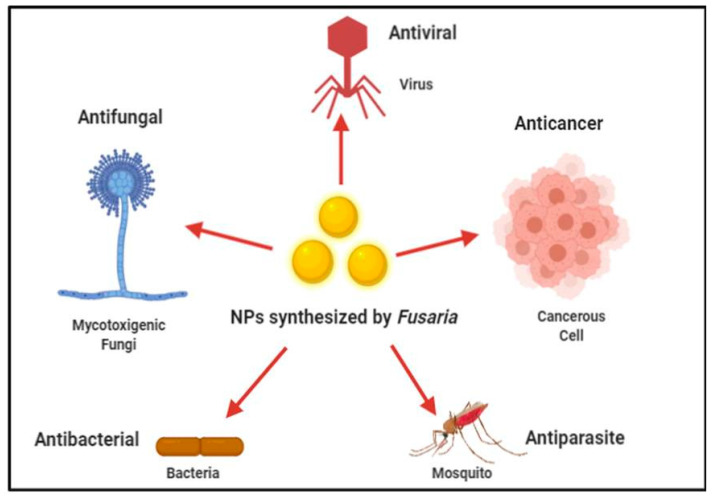
Biomedical applications of biogenic NPs synthesised from different *Fusaria*.

**Figure 7 jof-07-00139-f007:**
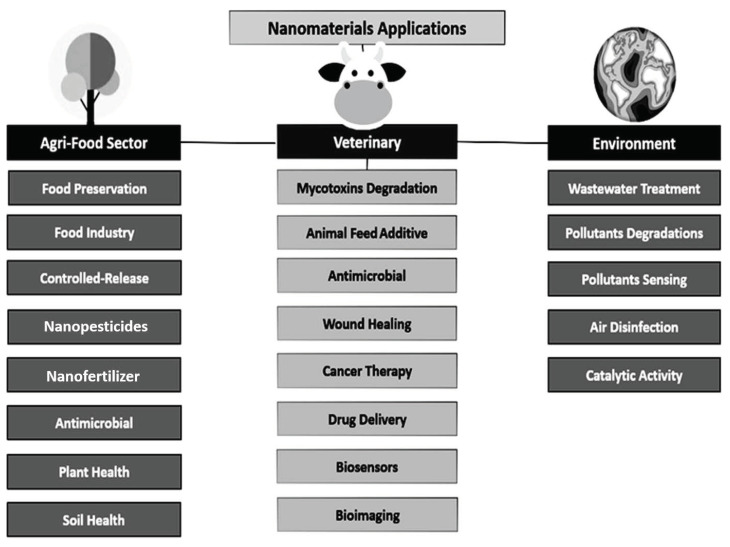
Applications of Nanotechnology in agri-food, veterinary medicine, and environment.

**Table 1 jof-07-00139-t001:** List of *Fusarium* spp. synthesizing metal Nanoparticles.

*Fusarium* spp.	Type of Nanoparticles	Reference
*Fusarium acuminatum*	Silver,	Ingle et al. [10]
*Fusarium solani*	Silver	Ingle et al. [11]
*Fusarium semitectum*	Silver	Basavaraja et al. [19]
*Fusarium acuminatum*	Gold	Tidke et al. [27]
*Fusarium culmorum*	Silver	Bawaskar et al. [28]
*Fusarium chlamydosporum* NG30	Silver	Khalil et al. [29]
*Fusarium equiseti, Fusarium tricinctum*	Silver	Gaikwad et al. [30]
*Fusarium proliferatum*	Silver	Gaikwad et al. [30]
*Fusarium keratoplasticum A1-3*	Zinc oxide Silver	Mohamed et al. [31] Mohamed et al. [32]
*Fusarium monoliforme*	Silver	Gaikwad et al. [30]
*Fusarium oxysporum*	Silver, Gold, Lead and Cadmium Carbonate, Strontium Carbonate, Cadmium Sulfide, Silica and Titania, Silica, Barium Titanate, Zirconia Platinum Magnetite CdSe Quantum dot CdTe Quantum dot Titanium oxide Chitosan Zinc Sulfide	Birla et al. [33] Bansod et al. [34] Sanyal et al. [35] Rautaray et al. [36] Ahmad et al. [37] Bansal et al. [38] Bansal et al. [39] Bansal et al. [40] Gupta and Chundawat [41] Bharde et al. [42] Kumar et al. [43] Senapati et al. [44] Ganpathy and Siva [45] Boruah and Dutta [46] Mirzadeh et al. [47]
*Fusarium oxysporum* f. sp. *cubense* JT1	Gold	Thakker et al. [48]
*Fusarium oxysporum 405*	Silver	Rajput et al. [49]
*Fusarium oxysporum* f. sp. *lycopersici*	Platinum Cadmium Sulphide	Riddin et al. [50] Cardenas et al. [51]
*Fusarium oxysporum* PTCC 5115	Silver	Karbasian et al. [52]
*Fusarium graminearum*	Silver	Ajah et al. [53]
*Fusarium scirpi*	Silver	Rodríguez-Serrano et al. [54]
*Fusarium semitectum*	Gold and Gold-silver alloy Silver Selenium	Sawle et al. [55] Madakka et al. [56] Abbas and Baker [57]
*Fusarium semitectum* (KSU-4)	Silver	Mahmoud et al. [58]
*Fusarium solani*	Zirconium Oxide	Kavitha et al. [59]
*Fusarium solani* ATLOY–8	Gold	Clarance et al. [60]
*Fusarium verticillioides*	Silver	Mekkawy Mekkawy et al. [61]

**Table 2 jof-07-00139-t002:** *Fusarium* isolates used in the green synthesis of different NPs.

Name of Fungi	Synthesised NPs	Localization	Size (in nm) and Shape	References
*Fusarium oxysporum*	Ag	Extracellular	10–20 and Spherical	Birla et al. [33]
			25–50 and almost spherical	Korbekandi et al. [108]
			-	Ishida et al. [76]
			5–13 and Spherical	Husseiny et al. [79]
			23 and Spherical	Hamedi et al. [109]
*Fusarium verticillioides*	Ag	Extracellular	-	Mekkawy et al. [61]
*Fusarium semitectum*	Au	-	25 and Spherical	Sawle et al. [54]
*Fusarium oxysporum*	Au	-	2–50 and Spherical, monodispered	Zhang et al. [110]
*Fusarium oxysporum*	Au-Ag bimetallic	Extracellular	8–14 and Quasi-spherical	Senapati et al. [111]
*Fusarium semitectum*	Au-Ag alloy	-	25 and Spherical	Sawle et al. [55]
*Fusarium oxysporum*	Fe_3_O_4_	Extracellular	20–50 and Irregular, quasi-spherical	Bharde et al. [42]
*Fusarium oxysporum*	Pt	-	70–180 and Rectangular, triangular, spherical and aggregates	Govender et al. [112]
*Fusarium oxysporum* f. sp. *lycopersici*	Pt	Extra-and intracellular	10–100 and Hexagonal, pentagonal, circular, squares, rectangles	Riddin et al. [50]
*Fusarium* spp.	Zn	Intracellular	100–200 and Irregular, some spherical	Velmurugan et al. [113]

## Data Availability

Not applicable.

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
