# Peer review of "Fusarium as a Novel Fungus for the Synthesis of Nanoparticles: Mechanism and Applications"

_jof, 2021, doi:10.3390/jof7020139_

Round 1

Reviewer 1 Report

In this paper, the authors have reviewed the synthesis of nanoparticles by Fusarium spp. and the application of NPs in different fields. The diversity of Fusarium species used for production and mechanism of NP synthesis are reported. Biomedical and agricultural applications are detailed. Finally, the toxicity of NPs is discussed. Overall the review offer a general overview about NPs. Nevertheless, the paper is more focused on NPs itself than on the producing fungi. There are some repetitions in the text (e.g. lines 308-313 already discussed in the chapter 2 "Mechanism of NP synthesis") that could be avoided.

Author Response

Point 1: There are some repetitions in the text (e.g. lines 308-313 already discussed in the chapter 2 "Mechanism of NP synthesis") that could be avoided. In this paper, the authors have reviewed the synthesis of nanoparticles by Fusarium spp. and the application of NPs in different fields. The diversity of Fusarium species used for production and mechanism of NP synthesis are reported. Biomedical and agricultural applications are detailed. Finally, the toxicity of NPs is discussed. Overall the review offer a general overview about NPs. Nevertheless, the paper is more focused on NPs itself than on the producing fungi.

Response 1: The repetitions have been deleted as suggested.

 The review is more focused on NPs than the producing fungi because the  guest issue is dedicated to “Fungal Nanotechnology

Reviewer 2 Report

The manuscript titles 'Fusarium as a novel fungus for the synthesis of nanoparticles: mechanism and applications' provides a good outlook of utilising the natural fungi for industrial purposes in an environmentally sustainable manner.  The manuscript is generally well written, and provides a great insight into this field. There are minor issues which need to be addressed before it can be accepted for publication. These are provided below.

Specific comments:

Line 4, Author name: Formatting error. The name should be "Mahendra Rai1,2", and not "Mahendra Rai1’2". (comma instead of apostrophe)

Line 72: Replace "In the current era, fungus plays..." by "Fungi play...". The term 'in the current era' is obsolete and does not add meaning to the article.

Lines 76-84: Provide atleast 1-2 references for each scenario. (antimicrobials, biological processing, textile, leather, paper and pulp, etc.)

Line 86: Replace "the genus Fusarium was used by..." by "the genus Fusarium has been used by..."

Line 89: Figure 1, The colour coding needs improvement as the current contrast is very difficult to read. Just keep a simple black-white format for this part, or keep the colour coding of boxes limited to 3 colours, if absolutely necessary. Also, there seems to be a formatting error in the boxes as I see "squares with ? mark". Keep this image simple and compatible. (Use Microsoft powerpoint instead of complex software).

Line 94: Table 1, Please make sure that the font in the table is uniform. There are traces of 'Calibri' in the References column.

Line 132 and elsewhere: Correct the "60◦ and 80◦C". The degree sign must be superscript.

Figure 2.: The ruler in these individual images is not visible. Can it be made slightly larger?

Line 217-219: Provide 1-2 references

Line 246: The IC50 value's unit must be corrected to ug cm-3 (and not ug cm3). Also, the authors use mL in some places (Lines 433-437, for example) and cm3 here. Please make sure to keep the unit uniform throughout the manuscript. 

Figure 7: Colour coding for the boxes. Please select better colours or go for black-white-grey combination. Keep it simple and visually legible.

Table 2: Size and shape column,  either remove the unit 'nm' from 1st row or add 'nm' to every entry in the table.

Line 376-384: This feels like a repeat from earlier sections and is not required in this section. Just start from 2nd paragraph.

Line 476 and elsewhere: The authors use 'sp.' for species here, but use 'spp.' in other places. Please make this uniform across the article. 

Line 502 and elsewhere: When using acronyms such as ROS, indicate the full name on first use.

Line 520 and elsewhere: The term 'ppm mL-1' is incorrect. Its either ppm or mg mL-1 (or mg L-1, more correctly). Also, please make sure if the conversions are used correctly. the concentration of 0.312 mg mL-1 does not equate to 1.187 ppm (correct conversion should be 312 ppm). This discrepancy must be addressed as it has severe implications. The dosing difference between 1.187 ppm and 312 ppm is not only massive, but also makes the whole system unstable, impractical and physiologically harmful.

References: Some of the references have DOI in them while others do not. Please make sure that this is addressed as per the journal recommendation. 

Author Response

Title: Fusarium as a novel fungus for the synthesis of nanoparticles: mechanism and applications

ManuscriptID:jof-1056144    
Type of manuscript: Review

Response to Reviewer 2 comments

Queries/Comments

Responses

Reviewer #2:

The manuscript titles 'Fusarium as a novel fungus for the synthesis of nanoparticles: mechanism and applications' provides a good outlook of utilising the natural fungi for industrial purposes in an environmentally sustainable manner.  The manuscript is generally well written, and provides a great insight into this field. There are minor issues which need to be addressed before it can be accepted for publication. These are provided below.

Specific comments:

Line 4, Author name: Formatting error. The name should be "Mahendra Rai1,2", and not "Mahendra Rai1’2". (Comma instead of apostrophe).

We thank the reviewer for appreciating the mss

Author name formatting error has been corrected.

Line 72: Replace "In the current era, fungus plays..." by "Fungi play...". The term 'in the current era' is obsolete and does not add meaning to the article.

"In the current era, fungus plays..." by "Fungi play...". has been deleted.

Lines 76-84: Provide at least 1-2 references for each scenario. (antimicrobials, biological processing, textile, leather, paper and pulp, etc.)

Two References have been added as per Reviewer’s suggestion

Line 86: Replace "the genus Fusarium was used by..." by "the genus Fusarium has been used by..."

‘the genus Fusarium was used by’ has been replaced

Line 89: Figure 1, The colour coding needs improvement as the current contrast is very difficult to read. Just keep a simple black-white format for this part, or keep the colour coding of boxes limited to 3 colours, if absolutely necessary. Also, there seems to be a formatting error in the boxes as I see "squares with ? mark". Keep this image simple and compatible. (Use Microsoft powerpoint instead of complex software).

The colour coding for Figure 1 has been changed to black and white with 600 dpi

Line 94: Table 1, Please make sure that the font in the table is uniform. There are traces of 'Calibri' in the References column.

Table 1, the font is corrected to have uniformity

Line 132 and elsewhere: Correct the "60◦ and 80◦C". The degree sign must be superscript.

The degree sign has been changed as superscript.

Figure 2.: The ruler in these individual images is not visible. Can it be made slightly larger?

Figure 2 the ruler has been made slightly larger.

Line 217-219: Provide 1-2 references

One Reference is added in Line 217-219.

Line 246: The IC50 value's unit must be corrected to ug cm-3 (and not ug cm3). Also, the authors use mL in some places (Lines 433-437, for example) and cm3 here. Please make sure to keep the unit uniform throughout the manuscript. 

Line 246: The IC50 value's unit has been corrected to ug cm-3

All the unit are kept uniform throughout the manuscript as per Reviewer suggestion

e.g mL to ml

Figure 7: Colour coding for the boxes. Please select better colours or go for black-white-grey combination. Keep it simple and visually legible.

Figure 7: has been replaced with B & W figure in high resolution

Table 2: Size and shape column, either remove the unit 'nm' from 1st row or add 'nm' to every entry in the table.

Table 2: Size and shape column, deleted the unit 'nm' in 1st row

Line 376-384: This feels like a repeat from earlier sections and is not required in this section. Just start from 2nd paragraph.

This section has been removed as per suggestion and started it from second paragraph.

Line 476 and elsewhere: The authors use 'sp.' for species here, but use 'spp.' in other places. Please make this uniform across the article. 

As per rules of speciation, the 'spp.' Is used for multiple species ‘sp.’ Is used for a single species. That is why on some places spp. is used while on others ‘sp.’ Is used.

Line 502 and elsewhere: When using acronyms such as ROS, indicate the full name on first use.

The suggested ROS correction has be made.

Line 520 and elsewhere: The term 'ppm mL-1' is incorrect. Its either ppm or mg mL-1 (or mg L-1, more correctly). Also, please make sure if the conversions are used correctly. the concentration of 0.312 mg mL-1 does not equate to 1.187 ppm (correct conversion should be 312 ppm). This discrepancy must be addressed as it has severe implications. The dosing difference between 1.187 ppm and 312 ppm is not only massive, but also makes the whole system unstable, impractical and physiologically harmful.

ppm mL-1 corrected to   ppm

References: Some of the references have DOI in them while others do not. Please make sure that this is addressed as per the journal recommendation. 

DOI for all references are added throughout review as per journal recommendation.

Reviewer 3 Report

This paper is a review for the synthesis of nanoparticles by fungus, in particular Fusarium. It has originality and overall presentation is good. However, there are some unclear points

Main concerns:

1 In the abstract is write that the review is  focused on the synthesis of silver nanoparticles and their use in different fields but reading the article often refers to nanoparticles in general or to nanoparticles different from silver. The review is focused on AgNPs or generic Nanoparticles? Please correct

2 The section 2.1.2 is not clear please rewrite

3 in the paragraph 3 the authors comment on the application of different metal nanoparticles in agriculture, the paragraph is too long and not very focused on silver. please simplify and better comment on the use of silver nanoparticles in agriculture

3 In the text (in particular in different paragraphs)  there are a lot of repetitions on the properties of nanoparticles. Please write once in the introduction

4 In the introduction in different points is write “etc”, please remove

Author Response

Responses to Reviewer 3 Comments

Comments and Suggestions for Authors

This paper is a review for the synthesis of nanoparticles by fungus, in particular Fusarium. It has originality and overall presentation is good. However, there are some unclear points

Thanks for appreciation

Point1: In the abstract is write that the review is  focused on the synthesis of silver nanoparticles and their use in different fields but reading the article often refers to nanoparticles in general or to nanoparticles different from silver. The review is focused on AgNPs or generic Nanoparticles? Please correct

Response1: The review is focused on general Nanoparticles. The suggested change in abstract has been made.

Point 2: The section 2.1.2 is not clear please rewrite.

Response 2: Section 2.1.2 has been re-constructed.

Point3: In the paragraph 3 the authors comment on the application of different metal nanoparticles in agriculture, the paragraph is too long and not very focused on silver. please simplify and better comment on the use of silver nanoparticles in agriculture.

Response 3: The part Application in Agriculture is simplified and reduced as per suggestion.

Point 4: In the text (in particular in different paragraphs) there are a lot of repetitions on the properties of nanoparticles. Please write once in the introduction

Response 4: Properties of nanoparticles mentioned in Introduction as per suggestion.

Point 5: In the introduction in different points is write “etc”, please remove.

Response 5: In the introduction etc removed as per suggestion.

Round 2

Reviewer 1 Report

The manuscript has been modified according to the reviewer's comments.

Reviewer 3 Report

After checking the changes authors have made, I am pleased to recommend the revised manuscript for publication in its current form